# Fundamental aspects to localize self-catalyzed III-V nanowires on silicon

J. Vukajlovic-Plestina[1], W. Kim[1], L. Ghisalberti[1,2], G. Varnavides[2], G. Tütüncuoglu[1], H. Potts [1], M. Friedl[1], L. Güniat[1], W.C. Carter[1,2], V.G. Dubrovskii[3] & A. Fontcuberta i Morral [1,4]

III-V semiconductor nanowires deterministically placed on top of silicon electronic platform would open many avenues in silicon-based photonics, quantum technologies and energy harvesting. For this to become a reality, gold-free site-selected growth is necessary. Here, we propose a mechanism which gives a clear route for maximizing the nanowire yield in the self-catalyzed growth fashion. It is widely accepted that growth of nanowires occurs on a layer-by-layer basis, starting at the triple-phase line. Contrary to common understanding, we find that vertical growth of nanowires starts at the oxide-substrate line interface, forming a ring-like structure several layers thick. This is granted by optimizing the diameter/height aspect ratio and cylindrical symmetry of holes, which impacts the diffusion flux of the group V element through the well-positioned group III droplet. This work provides clear grounds for realistic integration of III-Vs on silicon and for the organized growth of nanowires in other material systems.

[1] Laboratory of Semiconductor Materials, Institute of Materials, EPFL, 1015 Lausanne, Switzerland. [2] Departments of Materials Science and Engineering, MIT, Cambridge, MA 02139, USA. [3] ITMO University, Kronverkskiy Prospekt 49, St Petersburg, Russia 197101. [4] Institute of Physics, EPFL, 1015 Lausanne, Switzerland. These authors contributed equally: J. Vukajlovic-Plestina, W. Kim. Correspondence and requests for materials should be addressed to A.F.i M. (email: anna.fontcuberta-morral@epfl.ch)

Integration of compound semiconductors on silicon has been the holy grail of epitaxy and optoelectronics in the last 40 years[1,2]. Combining these two families of semiconductors would add optical functionality to the existing silicon electronics platform. Lattice and polarity mismatch remain the most challenging bottlenecks that result in detrimental dislocations and anti-phase boundaries in planar compounds. Epitaxy at the nanoscale through the formation of nanowires has been shown to circumvent these issues[3]. Semiconductor nanowires are filamentary crystals with a tailored diameter between few and hundred nanometers. It is thanks to their reduced diameter that anti-phase boundaries can be extinguished and dislocations either completely suppressed or reduced to misfit defects at the interface with the substrate, with minimal impact on the functional properties[4].

Among the methods to integrate compound semiconductor nanowires on silicon is the vapor–liquid–solid (VLS) growth, whereby solid nanowires precipitate from liquid droplets, supersaturated with the vapor phase precursors. The most commonly used external catalyst for the VLS growth is gold, which is unfortunately incompatible with silicon technology[5]. As an alternative, self-catalyzed (or self-assisted) growth arises as the gold-free VLS method fully compatible with silicon platform[6–10]. One well-known example of self-catalyzed VLS growth is gallium-assisted growth of GaAs nanowires by molecular beam epitaxy (MBE). Here, a gallium nanodroplet is used instead of gold to gather arsenic precursors to precipitate GaAs underneath[11]. Especially for the growth on silicon, preparation of gallium droplets turns out to be the key for a successful process[12,13].

The initial stage of growth, that is, the moment when the initial seeds of the nanowire are formed, is known to be crucial for the successful integration of nanowires on foreign substrates such as silicon[6,12–14]. In situ transmission electron microscopy (TEM) studies of nanowires as they grow has overturned our picture of the entire VLS process and led to its improvement[15–17]. In situ TEM studies at the interface with a crystalline substrate are extremely challenging and, to the best of our knowledge, inexistent.

Current understanding of nanowire growth is consistent with a layer-by-layer kind of process[18–20]. In this work, we show that quite surprisingly, formation of the nanowire growth seeds proceeds in a multiple layer fashion adopting unexpected configurations. Our work provides a clear guide for the optimal initiation of GaAs nanowire growth on silicon. The findings are general and thus provide the base for the successful integration of a wide range of compound semiconductors on silicon platform. The proposed paradigm shift has implications in the current understanding of nanowire growth in general.

## Results

**Optimal pattern design for a high yield**. We start by showing how the high vertical yield of deterministically positioned vertical GaAs nanowires can be obtained on silicon avoiding gold as the growth catalyst. While few groups have already achieved this goal in a certain parameter space, the fundamental experimental aspects for the successful nanowire growth remain in largest measure unformulated. We believe that the least known step is the formation of GaAs crystals within the lithographically defined holes in silicon oxide, filled with liquid gallium. Apart from the extreme cleanliness of the substrate achieved in state-of-the-art cleanroom facilities, neither the oxide thickness $h$ nor the hole diameter $d$ seems to be the key in a unique way. We claim here that it is rather the hole aspect ratio $d/h$ that has a prominent role in producing the best nanowire seeds and hence achieving the high yield of

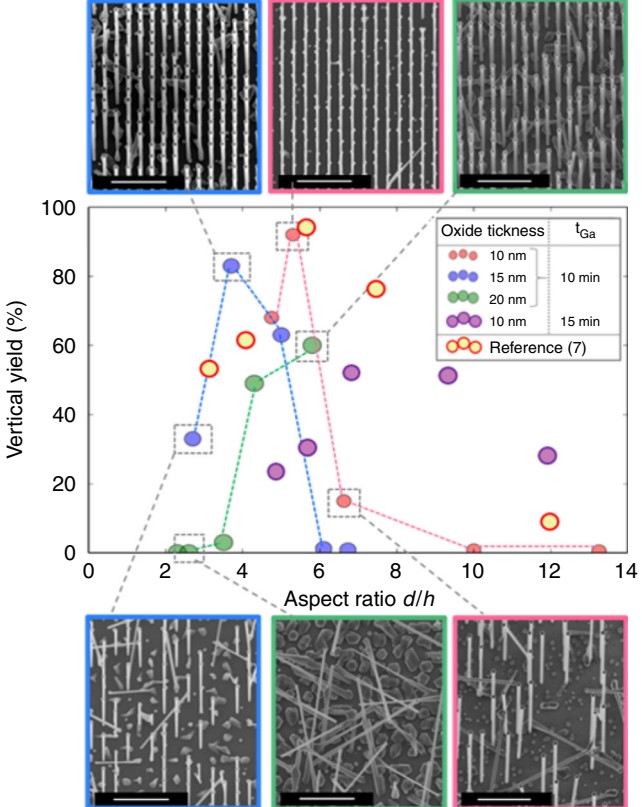

**Fig. 1** Optimization of the yield of vertical GaAs nanowires by the hole aspect ratio. Different values of the aspect ratio $d/h$ were obtained using three different oxide thicknesses $h$ of 10, 15, and 20 nm (corresponding to the red, blue, and green data points, respectively), and different diameters $d$. The hole arrangement and atomic force microscopy (AFM) profile measurements of the holes can be found in Supplementary Note 1. All samples were grown under the same conditions (details in supplementary material, Supplementary Figs. 1, 2). 20° tilted SEMs shown in the inserts illustrate the samples with the best and worst yields for each $h$ where the nanowires were formed (yield that equals zero for the case of $h = 15$ and 20 nm only parasitic growth was formed); color code of the insert frames is the same as for the data points. Scale bars correspond to 2 μm. Supplementary Note 2 gives the yield versus the aspect ratio over the full arrays. The maximum yield is obtained for the aspect ratios between 4 and 6. The yellow points represent the results of ref. [7], following the same trend

nanowires. This finding is summarized in Fig. 1, where we also include the results from another group[7] for completeness. The figure shows the statistics of vertical yield for GaAs nanowire arrays grown on silicon as a function of the $d/h$ ratio for different thermal oxide thicknesses $h$ (10, 15, and 20 nm), along with the representative scanning electron micrographs (SEMs). The top and bottom rows correspond to SEMs representing the best and worst yield, respectively. The gallium pre-deposition time was fixed at 10 min. The substrate preparation and growth conditions for all our samples were identical, with the details given in the supplementary material (Supplementary Figures 1 and 2). Clearly, the yield of vertical nanowires reaches its maximum for the $d/h$ ratios between 4 and 6 for all three oxide thicknesses used in this work. The same trend is observed for nanowires reported in ref. [7]. The morphology and size of the holes are only relevant in the initial stages of growth where the nanowire growth begins. These results suggest a fundamental mechanism occurring at these stages.

**Droplet positioning**. One aspect to explain the improvement of the yield with the aspect ratio could be the preparation of growth by correctly filling the holes with gallium. For each given pre-deposition condition there would be an optimal diameter for complete filling of holes. The latter may indeed favor homogeneous contact angle of the gallium droplets close to 90° with the substrate, because the holes exhibit vertical walls. It has been shown that a 90° contact angle is optimal for orienting the nanowires perpendicularly to the substrate[12]. We monitored the gallium droplet formation inside the holes as a function of time, the step that we call pre-deposition. Pre-deposition of gallium in the holes has previously been reported as influencing the final yield[7,8,13]. Going back to Fig. 1, one can see that larger holes and thicker oxide (but still with the optimal aspect ratios between 4 and 6) never reached the best yield of 90%. Hence, one can infer that the yield in larger diameter holes (for example, 75 and 90 nm) should be improved simply by a longer gallium predisposition to ensure their filling. By increasing the gallium pre-deposition time from 10 to 15 min, we can indeed increase the yield in the larger diameter holes. In some cases the yield increases from almost zero up to 50%, which is a significant improvement. Still, as the yield does not improve beyond 50%, it indicates that other conditions need to be fulfilled.

Atomic force microscopy (AFM) measurements of the gallium droplets as a function of the filling time were performed ex situ at room temperature. The results for the 45 nm diameter holes are illustrated in Fig. 2a. The droplet always starts at the edge of the hole. For surface energetic reasons, it first pins at the oxide-substrate interface line and grows from there toward the other end of the hole. For this geometry of the hole and deposition conditions, we are able to achieve symmetric gallium droplets with a contact angle of ~ 90° with 7 min of pre-deposition time. Shorter pre-deposition leads to incomplete filling of the holes. For

longer pre-deposition, the droplet swells by increasing the contact angle larger than 90° and finally starts crawling out of the hole onto the oxide surface.

To better understand the filling process, we computed the evolution of the droplet filling the holes using Surface Evolver[21] (the details of these calculations are given in the supplementary material, Supplementary Figs. 10, 11, 12). We model the equilibrium shape of liquid gallium in the constrained configuration, given by the geometry, droplet volume and relative surface energies of different interfaces. Figure 2b depicts the evolution of the droplet equilibrium morphology as a function of its volume, in a $SiO_2$ cylindrical cavity with a fixed volume $V_0$ and an aspect ratio of 4 on a Si(111) substrate. The droplet nucleates at the edge of the hole to minimize the total surface energy, because the interface with the mask has a lower energy than with the vapor. After that, the droplet evolves asymmetrically from the edge by filling a part of the hole. Symmetrical droplet shapes are achieved only by the complete filling of the hole. For larger droplet volumes, they continue to fill the cavity in a symmetric way and by increasingly wetting the sidewalls. The conditions to achieve symmetric filling of the hole depend on both the size of the hole and the volume of the droplet, the latter controlled by the pre-deposition time. Intuitively, complete filling of the holes may lead to an improvement of the yield. One may think that this should naturally lead to homogeneous layer-by-layer growth at the bottom of the hole.

**Ring-like nucleation at the mask-substrate interface line**. The fundamental mechanisms affecting the vertical yield should be related to the very initial stages of growth down to the first few monolayers. To gain access to this stage, we characterized the seeds below the gallium droplets, formed in the initial stage of growth after the gallium pre-deposition. We have recently shown that the incubation time required to start the nanowires can be as long as a few minutes[22]. We have also found that the distribution of the incubation times can be quite broad for low degrees of supersaturation in the vapor phase[22]. This broad distribution gives a more representative picture of possible initial configurations of the nanowire growth seeds. These results are consistent for all samples investigated. Time-series have been conducted, and all the samples displaying growth times that are comparable to the incubation stage were showing identical behaviors. We have thus chosen the growth conditions yielding a broad distribution of the starting times for nanowire growth, so that the evolution of seeds can be observed in a single sample and in a consistent manner.

Figure 3a–c shows the representative SEM and AFM images of the arrays grown for 2 min. We selectively removed the gallium droplets by dipping the sample in 37% HCl for 15 min. This enables us to see the first nanowire layers inside the holes, previously hidden by the droplet. We present the results for the arrays corresponding to the best and worst yields, for the holes of 45 nm diameter (close to 90% yield) and 90 nm diameter (almost zero yield), at an oxide thickness of 10 nm.

Removing the droplet reveals the surprising shapes of GaAs seed crystals. Namely, we observe multi-layer terraces and rings starting at the Si/SiO$_2$ interface instead of a more expected single layer structures. A clearer view can be found in the magnified images in Fig. 3c. We observe more regular rings in the optimized sample with 45 nm holes. Conversely, the seed shapes turn to be extremely different in the bad sample with 90 nm holes. Here, the growth starts with clearly faceted islands, evolving randomly at different points of the oxide-substrate interface line and propagating toward the other side of the hole. These seeds remind of the crystal shapes observed in the initial growth stage

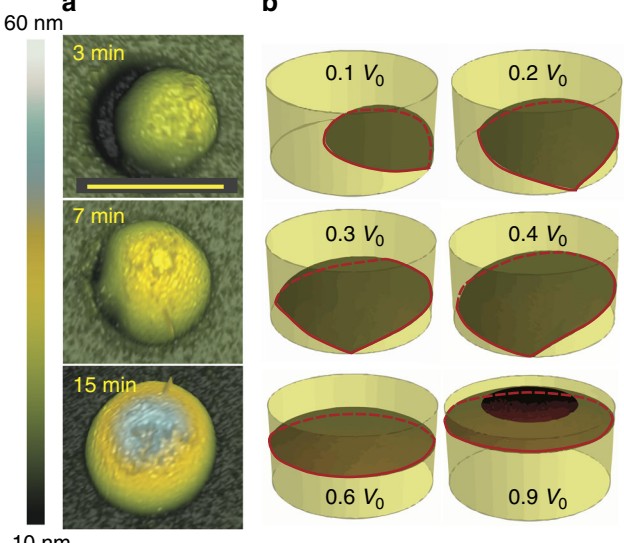

**Fig. 2** Filling of the nanoscale holes in SiO$_2$ mask on silicon. **a** AFM images of the gallium droplets in a 45 nm diameter hole for three different pre-deposition times. Scale bar corresponds to 50 nm. **b** Illustrations of the evolution of the equilibrium droplet shape for increasing values of their volume. $V_0$ is the total volume of the hole. At small volumes, the droplet starts wetting asymmetrically from one side of the hole until the bottom interface is in full contact with the liquid phase. Further filling proceeds by increasing the wetting at the sidewalls. The morphologies were obtained using Surface Evolver for sessile droplets constrained in a cylindrical cavity with an aspect ratio of 4

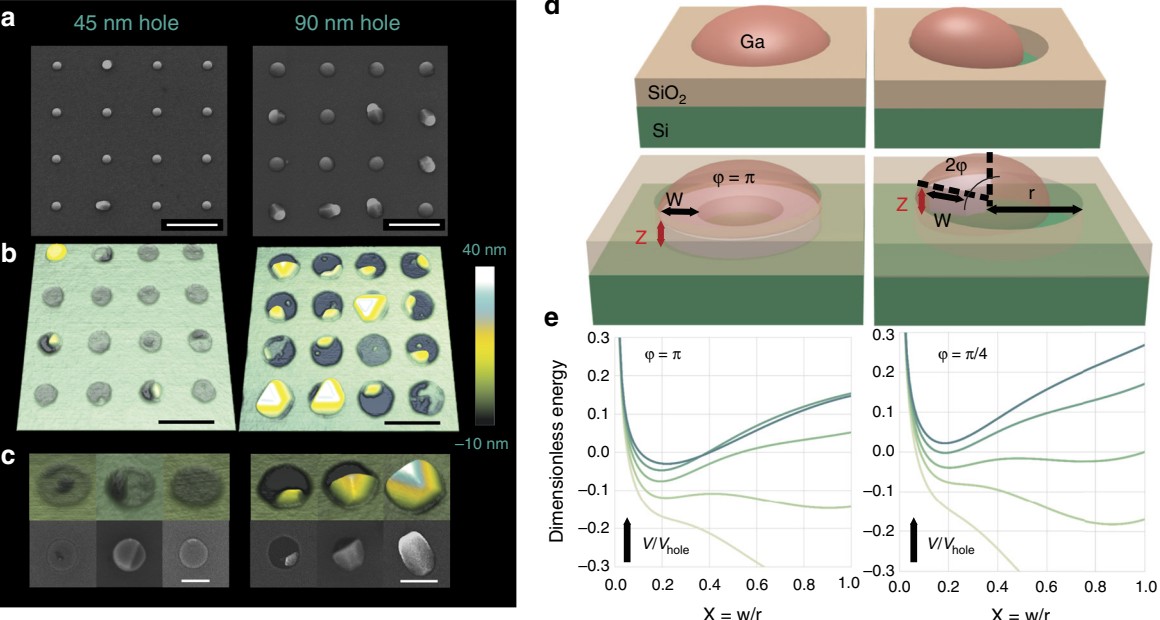

**Fig. 3** Seeding the nanowire growth. **a** SEM images of 400 nm pitch arrays grown for 2 min in 45 and 90 nm holes. Scale bar is 400 nm. **b** AFM images of GaAs crystals in the holes revealed after the droplet removal. The Images were taken from the same sample as in **a**, but over 200 nm pitch area. Scale bar is 200 nm. **c** Closer view of the structures shown in **b**. Scale bars are 100 nm (left), 200 nm (right). **d** Two different configuration of the droplet within the opening—hole fully filled and partially filled and bellow illustration of GaAs crystals of height $z$ (in red) forming a step ($\varphi < \pi$) or full ring ($\varphi = \pi$) underneath the large and small gallium droplets. **e** Free energy of forming a GaAs crystal for the plausible parameters of the system. Each curve on the plots corresponds to the different values of relative volumes $V/V_{hole}$ that is measured in the units of $V_{hole} = 2\pi r^3$. The preferred width of the crystal equals $r$ only for small volumes of deposited GaAs (the curve at $V/V_{hole} = 0.02$). After that, the local minimum of free energy develops and stabilizes at ~0.2$r$, corresponding to the formation of 3D ring-like or stepwise structures. The arrows on the plots indicate the increase of value of the relative volume. The values of $V/V_{hole}$ are: 0.015, 0.02, 0,03, 0.05, 0.1 for $\varphi = \pi$ and 0.003, 0.004, 0.005, 0.007, 0.01 for $\varphi = \pi/4$

of GaAs nanowires tilted by three-dimensional (3D) twinning effect[6].

Nucleation of GaAs crystals at the bottom of the hole in the form of step or ring having heights much larger than monolayer (additional AFM profiles of these structures are presented in the supporting material, Supplementary Fig. 7) is highly unexpected and calls for discussion. Indeed, for the standard VLS growth of developed nanowires far away from the substrate, theoretical considerations[23–26], and in situ growth monitoring[16,17] reveal layer-by-layer growth so that the flowing steps advance on a single layer basis[27]. However, growth within the holes is different —first, GaAs crystal forms on the lattice mismatched silicon substrate and, second, the crystal has lateral solid–solid interface with the SiO$_2$ mask rather than free sidewalls in contact with vapor. So far, in situ investigations of the initial stages of growth on a patterned substrate have not been achieved.

To describe the multiple layer height of the nanowire seed within the hole, we developed a model for which the full description is given in the supplementary material, Supplementary Figs. 8 and 9. Here, we just describe its main ingredients. We calculate the free energy of forming a GaAs crystal at the bottom of the hole below the gallium droplet and restricted by vertical walls of the oxide mask. This energy contains contributions from the surface energies of different interfaces[22] and, very importantly, the strain-induced term provoked by the lattice mismatch between GaAs and the underlying silicon substrate[3,28–30]. We consider both homogeneous ring-like and localized shape, described by the angle $\varphi$ and width $w$, as depicted in Fig. 3d. Within the model, the $\varphi$ value is determined by the filling factor of the initial gallium droplet in the hole as described above.

We then calculate the free energy as a function of the $w/r$ ratio for different volumes of GaAs (with $r = d/2$ as the hole radius), as

shown in Fig. 3e. Clearly, $w/r = 0$ corresponds to growth along the SiO$_2$ edge, while for $w/r = 1$ layer-by-layer growth becomes more favorable. The energetically preferred $w/r$ is determined by the free energy minimum. The graphs in Fig. 3e show that layer-by-layer growth at $w/r = 1$ is preferred only for very small volumes of deposited GaAs (typically <5% of the hole volume), regardless of $\varphi$. For larger volumes of GaAs, the growth seeds tend to nucleate as 3D structures at the hole edges, with the optimum $w/r$ saturating at ~0.2 for these parameters. Localized nucleation results in the formation of a nano-crystallite on a corner. The formation of nano-facets in the case of incomplete filling facilitates 3D twinning and therefore a tilted growth[6]. On the contrary, extended nucleation in the form of full ring at the oxide-substrate interface line in the case of the complete filling allows for the formation of regular GaAs seed and further vertical growth of nanowires. Transformation from layer-by-layer to 3D growth after exceeding a certain critical deposition thickness is strongly reminiscent of the Stranski–Krastanow islands;[31] however, here 3D growth occurs in different morphology.

**Diffusion field supporting ring-like nucleation profiles**. While the above model explains the energetically preferred shape of the nanowire seeds and the importance of symmetrically filing the holes, it does not yet clarify the role of the $d/h$ ratio. We now consider an additional effect which, to the best of our knowledge, has not been studied so far—the role of a gradient in the group V species in the droplet (As$_4$ in our case). Figure 4a illustrates the configuration of the gallium droplet in the symmetrically filled hole and the effect of the directionality of the As$_4$ flux in MBE[32]. As adatom, arsenic does not diffuse on the surface. If it is not incorporated it is further desorbed[33]. In our machine, the molecular flux impinges on the surface at an angle of 45°. The arsenic flux into the

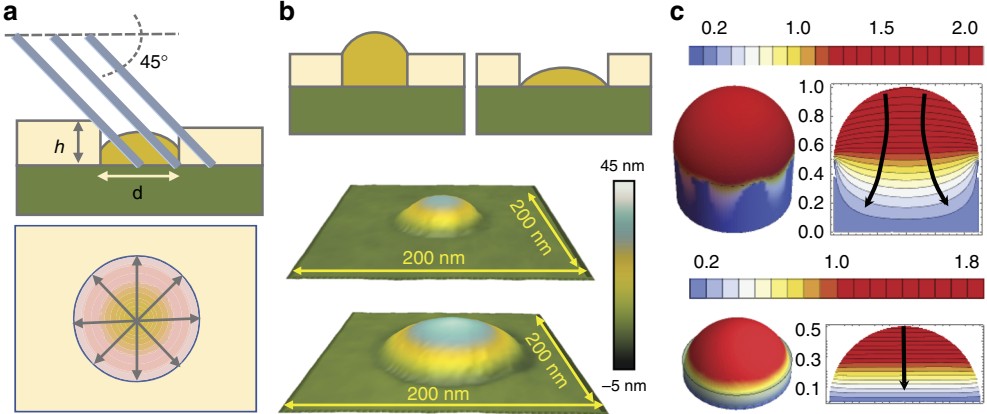

**Fig. 4** The diffusion model explaining the role of the hole aspect ratio $d/h$ in improving the yield. **a** Top: schematics of the droplet configuration within the opening with respect to the impinging arsenic flux; bottom: schematics of the arsenic gradient created due to the sidewall shadowing. **b** AFM measurements of gallium droplets for 10 min pre-deposition time at an oxide thickness of 10 nm, showing the modification of the droplet shape from full wetting, through partial wetting of the mask walls and ultimately non-wetting of the mask walls for largest $d/h$ ratio; top sketches showing the two wetting configurations in the cross-section **c** 3D representations and 3D cuts of the calculated arsenic concentration profiles for the two wetting configurations corresponding to different $d/h$ (top: $d/h = 2$, bottom: $d/h = 6$). The values in the color scale correspond to the arsenic concentration (a.u.)

gallium droplet is thus shadowed by the oxide mask at the interface with the substrate. The concentration of arsenic dissolved in the gallium droplet should thus be inhomogeneous, creating a gradient toward the substrate. The aspect ratio of the hole affects the shadowing and thus the direction of the gradient of dissolved arsenic in the droplet. Figure 4b shows AFM measurements of gallium droplets filling 30 and 90 nm diameter holes in a 10 nm-thick oxide mask, obtained after 10 min pre-deposition. These two configurations lead to the nanowire vertical yields of 65% and 3%, respectively (optimal yield is obtained for 45 nm holes, wetting configuration similar to 30 nm). Both droplets sit symmetrically in the nanoscale hole. However, for the small $d/h$ ratio, the gallium droplet is pinned on top of the hole, exhibiting a contact angle close to 90°. For the high $d/h$ ratio, the droplet is pinned at the bottom of the hole, wetting very little the sidewalls. We calculated the arsenic concentration profile for these configurations (the intermediate configurations are given in the supporting material, Supplementary Fig. 11), by numerically solving Fick's equation, taking into account the sample rotation in MBE. Figure 4c depicts the steady state concentration profiles in droplets with the same $d/h$ aspect ratio and configuration as measured by AFM. We present both the 3D plot and a cut through the droplet center. Different droplet colors illustrate the gradient of the arsenic concentration and thus the direction of its diffusion, also indicated by arrows in the cross-sectional plots (right column of the figures). For low $d/h$ ratios, the arsenic flux is directed towards the oxide-substrate interface line. This results in a strong shadowing effect in the holes with low $d/h$ values. The lack of arsenic in these zones creates a permanent concentration gradient. By increasing the $d/h$ ratio, the arsenic diffusion flux is turned outward from the substrate-oxide interface line. When the $d/h$ ratios are too large, concentration gradient toward the oxide-substrate line becomes negligible. Hence, the arsenic diffusion towards the substrate becomes homogeneous throughout the hole.

Let us now discuss the relevance of the effect using some numerical estimates. The diffusion coefficient of arsenic atoms in liquid gallium at the growth temperature was estimated at $2 \times 10^{-12}$ m²/s[34]. Therefore, the characteristic time for arsenic to diffuse through the droplet is about $5 \times 10^{-2}$ s. This is much smaller than the characteristic time of the single layer formation in the steady state nanowire growth (~1 s)[17,27], while the nanowire

nucleation delay is much longer and may take a few minutes[35]. As expected, diffusion of arsenic cannot be the limiting factor for the nanowire growth or nucleation. However, it is expected that the diffusion profile influences the location of nanowire nucleation and in this way contributes to the vertical yield. For the symmetrical gradient toward the hole periphery, the liquid wets the hole's bottom edge and thus nucleation will occur there. Subsequent growth will cause this nucleus to cover the bottom edge, as discussed above. On the contrary, a reduced arsenic flux toward the oxide-substrate interface line or asymmetric diffusion profiles in the case of asymmetric hole filling lead to a higher degree of tilted growth. Finally, we address the decrease of the yield for the smallest $d/h$ ratios. We believe that this decrease is mainly due to the difficulties in keeping symmetrical hole at small diameters. Given the strong gradient toward the oxide-substrate interface line at small $d/h$ ratios, any inhomogeneity in the hole circularity breaks the cylindrical symmetry of the flux. The decreased yield in this case is due to a higher arsenic gradient and its sensitivity to the hole asymmetry, which strongly increases for the smaller hole diameters (Supplementary Fig. 11). Precise nanofabrication of more symmetrical holes in the oxide layer is expected to increase the yield for the smallest $d/h$ values.

In conclusion, we have provided evidence of several factors that affect the yield of deterministically positioned GaAs nanowires obtained by the gallium-assisted method on silicon substrates. In particular, we have demonstrated the role of the droplet positioning, the hole aspect ratio and symmetry in connection with the flux of the dissolved arsenic towards the oxide/substrate interface line. We have elucidated the reason why the initial nanowire growth seeds emerge as 3D objects rather than growing in the layer-by-layer mode. All these details allowed us to optimize the yield of GaAs nanowires grown in regular arrays. We believe that these results open the way for the realistic integration of self-catalyzed III-V nanowires on silicon. They also provide original insights on the formation of nanowires and can be translated to other material systems that can be grown with low surface energy catalysts such as GaSb, GaP, InAs, InP, GeSn, and Si[36–41].

## Methods
**Patterns**. Si substrates with a thin layer of thermally grown silicon oxide were patterned with arrays of holes of different diameters. The nominal diameters of the

holes varied from 30 to 90 nm in increments of 15 nm (Fig. 1) while their different aspect ratios were achieved by using different oxide thicknesses: 10, 15, and 20 nm. The patterning process was realized by electron beam lithography followed by dry (reactive ion etching –RIE) and wet etching (buffered hydrofluoric acid—BHF). The last step in the substrate preparation before growth was the so-called "last dip". It refers to the 2 s dip in the BHF bath to remove the native oxide created at the bottom of the holes. The diameters of the holes, as with the oxide thickness, can differ with respect to the nominal diameters due to variations in the etching processes, especially in the "last dip".

**Nanowire growth.** All of our growth experiments were performed using the same material fluxes and the same growth temperature: gallium flux corresponding to the GaAs growth rate of 1 Å/s, the $As_4$ partial pressure of $2 \times 10^{-6}$ Torr and the substrate temperature of 635 °C as measured with the calibrated pyrometer. The Ga pre-deposition time was varied, same as the growth time.

**Characterization.** The substrates were thoroughly studied by AFM and spectroscopic ellipsometry (SE). AFM was used to precisely determine diameter of the holes, shape of the droplets and morphology of the nanowire seeds. The morphology and yield of the nanowire arrays and substrates were also investigated by scanning electron microscopy (SEM).

**Simulations/theory.** The morphological evolution of the droplet filling a cylindrical cavity formed by the $SiO_2$ sidewalls was computed through the software Surface Evolver. The equilibrium shape was found for different volumes of the droplet filling holes having aspect ratio equal to 2. The interfacial energies controlling the resulting final shape are implemented by the value of the contact angle, 51° for gallium on silicon and 116° for silicon oxide.

The nucleation model of GaAs nanocrystals from differently shaped gallium droplets in the holes is based on minimization of the free energy of the system for different droplet volumes. The free energy includes the surface energy terms originating from different interfaces and elastic energy contribution due to the lattice mismatch between GaAs and silicon at the bottom of the hole.

The concentration profile of the Arsenic diffusing into Gallium droplet was computed by solving the Fick's law by means of Finite Element Approach. The boundary conditions are the zero-diffusion through the substrate and the presence of an oriented impinging flux. The concentration profiles shown are the result of a time average done for rotating flux, in order to resemble the experimental conditions. The effect of the material shadowing is analyzed by changing the aspect ratio of the walls of the cavity.

## Data availability

The data sets generated and/or analyzed during the current study as well as the code are available on https://doi.org/10.5281/zenodo.2541732.

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

## Acknowledgements

EPFL authors thank SNSF funding through Grants Nos. 200021_169908 and IZLRZ2_163861 and the H2020 program through grant 'INDEED'. V.G.D. thanks the Ministry of Education and Science of the Russian Federation for financial support under Grant 14.587.21.0040 (Project ID RFMEFI58717X0040). We thank A. Giunto for fruitful discussions.

## Author contributions

J.V.-P. and W.K. performed substrate preparation and growths. J.V.-P. performed statistical analysis of yield and growth optimization. W.K. performed AFM analyses. L. Gh. and W.C.C. performed the surface evolver simulations. Lea G., G.V. and W.C.C. performed the diffusion calculations. V.G.D. performed the nucleation model of GaAs in the holes, in discussion with L. Gh., J.V.-P., W.K. and A.F.i.M. J.V.-P., W.K., G.T., H.P., M.F., L.G., V.G.D. and A.F.i.M. contributed to the discussions on the growth mechanisms. J.V.P. created the figures. AFiM proposed the idea, followed, and directed the work. The manuscript was mainly written by J.V.-P., V.G.D. and A.F.i.M. All authors have contributed to the correction and proofreading of the manuscript.

## Additional information

**Competing interests:** The authors declare no competing interests.

**Journal Peer Review Information**: *Nature Communications* thanks the anonymous reviewer(s) for their contribution to the peer review of this work. Peer reviewer reports are available.

