## [Peer Review File · Nature Communications]

Reviewers' comments:

Reviewer #1 (Remarks to the Author):

This paper addresses an important problem, that of achieving consistent vertical growth of III-V nanowires at chosen positions on a Si substrate. The authors start with the standard approach of growing in holes in an oxide mask, and they systematically examine the effect of hole dimensions and other factors, identifying the aspect ratio of the hole as the key factor determining successful growth. Figure 1 convincingly summarizes the main evidence. They then use modeling to gain a deeper understanding of why the aspect ratio is important. In this way they provide valuable new insight into the factors controlling growth.

Besides the technological importance of this advance, the progress toward a fundamental understanding of mask-controlled growth is fascinating. Given the quality and broad interest of the work and the importance of the results, the paper seems highly appropriate for Nature Communications. There are some points in the results and discussion that could be clarified prior to publication, and these are addressed below.

Figure 2 shows "AFM images of the gallium droplets". Is the AFM performed in situ at the growth temperature? This should be stated explicitly. If the AFM is done at room temperature or even in air, it would still give useful insight, but in any case the reader should be informed how far these measurements directly reflect the growth conditions, since the wetting could change somewhat with temperature and atmosphere.

The authors "claim here that it is rather the hole aspect ratio d/h that plays the universal role in producing the best nanowire seeds and hence achieving the high yield of nanowires." While the claim is adequately supported by the results, the wording seems a bit strong. The thickness ranges over only a factor of 2, while the aspect ratio of maximum yield varies by at least a factor of 1.5 (perhaps more, since we never see the curve for the 20nm film turn down). A more nuanced wording would seem appropriate, to reflect both the great advance in identifying the role of aspect ratio, and the remaining gaps in understanding and in achieving complete control.

Does the theoretical modeling support the idea that hole aspect ratio is the key? Perhaps the takeaway message from the modeling could be made clearer for the non-expert reader. It seems that the modeling can be viewed as two quite separate pieces. The first involves the wetting of the Ga droplet. The key message here is that the hole should not have too wide/shallow an aspect ratio. If it does, the droplet will grow over the edge before it wets the entire bottom. This sets an upper limit on d/h , but it doesn't set any lower limit. The results shown using Surface Evolver do not actually illustrate this failure mode, but I hope I have understood correctly, because the general principle seems simple and convincing. Since the wetting has no inherent length scale, it is truly the aspect ratio which matters here.

Then the puzzle becomes, why does a deep narrow hole not work? Here the modeling becomes more complex, and the inference more indirect. But the message seems to be that shadowing and As diffusion are causing failure for small d/h . The amount of shadowing is directly related to d/h , but diffusion is more complicated, so I'm not sure how directly this supports saying that d/h "plays the universal role". If the authors can offer any clarification or simple perspective, that would further improve the paper.

Some minor details:

I found Fig 2b a little confusing. I assume it is generated by Surface Evolver. Is the darker color a

projection of the droplet volume, as one might intuitively expect by analogy with TEM? It looks more like a projection of the liquid-vapor interface; but if that is the case, it might be helpful to add a color for the liquid-oxide interface.

The paragraph describing the Surface Evolver calculations ends with a sentence "Complete filling of the holes leads to an improvement of the yield (with the highest values obtained within the optimal d/h from 4 to 6)." As the concluding sentence of a paragraph about modeling, it sounds as if this is a result of the modeling, but I don't think the authors meant to suggest this. A few extra words could clarify the distinction between model results and experimental results, along with the relation between the last sentence and the rest of the paragraph.

Reviewer #2 (Remarks to the Author):

This manuscript discusses self-catalyzed growth of GaAs nanowires on Si and in particular focuses on obtaining good nucleation and yield. There are interesting aspects and the work may be suitable for this journal. However I found it very difficult to read and am unsure what the authors' central claim is, even after reading it a few times. In order to be suitable for publication, this manuscript needs to be rewritten somewhat so that it is clear what the main claim or breakthrough is, and how this is supported by the experimental data. I suggest the authors focus on clarifying this in a revised manuscript.

To elaborate, the abstract gives the impression that positioning of nanowires on Si, or perhaps yield of positioned nanowires on Si, is the important breakthrough in the manuscript. However after reading the manuscript I have the impression that this in fact has been demonstrated before (in the section Optimal pattern design the authors indicate that similar results have been shown in Ref 7 for example). The abstract also seems to indicate that a new understanding of the nucleation is presented. But this seems to be a small part of the paper and appears to be rather speculative. The actual major result of the paper needs to be clarified here and throughout the paper.

Some more specific points that should be addressed:

1. The abstract contains the sentence 'Contrary to common understanding, nucleation of nanowires occurs preferentially at the triple phase line between the liquid droplet...'. However it appears that preferential nucleation at the triple is exactly the current understanding in literature, so as written this sentence is factually incorrect. From the next sentence the authors appear to be distinguishing between the ring-like nucleation they discuss in the paper and more conventional nucleation followed by step flow, but this needs to be rephrased
2. The introduction contains a paragraph discussing in-situ investigations of nucleation in nanowires. From this discussion I expected to see this topic appear again in the paper, but it doesn't appear to be related. It would help if this paragraph were given some context to how it relates to the present work, which doesn't seem to have anything to do with in-situ TEM.
3. As a general note, the authors refer to III-V nanowire growth in the title and abstract and claim that the results are general, but it appears that only GaAs has been studied. Is there any actual evidence that this work will apply to other III-V nanowires, which from my understanding are much more difficult to grow with this method? In the absence of direct evidence I think the claims of generality are exaggerated and should be reduced.

4. The observations of early stages of growth after removal of Ga droplets were very interesting and I would suggest the authors put more emphasis on this part of the paper. However I am not entirely convinced of the authors' claim that one sample is sufficient to see all stages of nucleation/early growth. The claim is that since nucleation occurs at different times for different holes, all stages can be seen simultaneously. However it could also be that the shapes evolve differently for different holes, and therefore I would argue that the authors should still have considered samples with several growth times, at least to prove their argument that their interpretation of the results is generally correct (and counter the impression that much of their model is just speculation).

5. The last section, on diffusion in the droplet, appears to be purely a theoretical prediction. Can the authors comment on this, and how this could be tested experimentally (for example, would this model predict any trends that might be possible to observe?). Moreover, are there other possible explanations for the observed dependence on d/h of the holes, and how could these be tested against the proposed explanation?

Reviewer #3 (Remarks to the Author):

This is a very interesting paper focusing on an important challenge of epitaxial deposition of a II-V semiconductor (GaAs) on a silicon substrate. To alleviate the problems of lattice strain and mismatch in interface polarity, GaAs nanowires are deposited onto the substrates in specific, targeted positions using patterned holes in a silicon oxide mask. The authors study how the size of the holes--specifically, the aspect ratio between the hole diameter and the height of the oxide--contributes to the orientation of the nanowire growth--whether the nanowires grow vertically, or at some other random angle. The experiments are very well-done and the data convincingly show that this hole aspect ratio, and the initial deposition of Ga into the holes, is what determines the orientation of the nanowires. The authors propose a model for this based on how the nanowires nucleate in the holes (i.e., at the edges of the holes if the Ga droplets are not completely filling the holes) and how As flux can also lead to diffusion gradients in the seeding droplet under certain conditions. I think that the experimental data are solid, and the proposed model seems convincing to me. I recommend publication of the paper with some additional minor comments:

1. In the abstract, defining that the aspect ratio is the hole diameter to the mask oxide thickness (line 19, pg 1). I had to read through the paper to figure out what aspect ratio was being referred to in the abstract, and since this is key to the paper, it would be useful to define it in the abstract.

2. The authors refer to "layer-by-layer" growth and growth in a "multilayer" fashion often throughout the paper, yet never really define what layers they are referring to. I understand that researchers in the field will know, but the more general audience, even of materials researchers, probably will not know that they are referring to how the nanowires grow vertically from the substrate by the controlled addition of atomic layers... it would be helpful to have an explicit statement early in the paper about how the authors are defining layer-by-layer growth, etc.

3. Pg 2, Line 43...the sentence about the absence of in situ TEM of nanowires growing from a substrate was confusing to me... at first, I thought that the paper was going to show in situ TEM data of the nanowire growth, but did not. I would revise this somehow to perhaps just say that nobody knows how to do such an in situ TEM imaging, or something like that. At least clarifying that the paper is not going to show any in situ TEM data might be helpful to the reader.

4. Related to the data in figure 1, I imagine that there is also a critical hole diameter that above

which, nanowires do not grow epitaxially/vertically, regardless of the hole aspect ratio. Do the authors have some measure of this value, and can they state it for completeness? This is what the ref 3 is really about...the critical diameter needed to achieve a kind of defect-free growth. Also, is there a minimum diameter at which point the nanowires also do not grow? Or is the patterning dimension limited so this minimum hole size can't be reached? A comment about this lower limit would be useful as well.

5. Pg 5, line 91. Do the authors need to add a reference for the statement? "It has been shown that..." Or is this an introductory remark for the work that they are going to be talking about. It was confusing to me whether this was a sentence introducing the findings of this paper, or describing some previously published result.

6. Figure 2. It would be helpful to have some labels placed on the images in (b). Are these snapshots at different times in the growth? Or with more Ga deposited? Also, the caption should state that those are computed shapes with Surface Evolver. This is stated in the text, but the caption offers little detail about what is shown in the image. Also, the scale of the scale bar in (a) is not provided. (Is it 45 nm?)

7. It would be helpful to add a sentence or two about the MBE deposition process in terms of how the flux is directional at 45 deg. Maybe provide a reference to a review paper or something like that? I'm more familiar with selective area CVD processes. In those cases, there is a higher diffusion flux at the boundaries of the hole than in the interior because of the hole geometry. Can something like this also happen in the MBE process or are the reactants somehow being accelerated to the substrate and impinging onto the substrate. A very brief explanation of this would be useful.

Reviewer #1

This paper addresses an important problem, that of achieving consistent vertical growth of III-V nanowires at chosen positions on a Si substrate. The authors start with the standard approach of growing in holes in an oxide mask, and they systematically examine the effect of hole dimensions and other factors, identifying the aspect ratio of the hole as the key factor determining successful growth. Figure 1 convincingly summarizes the main evidence. They then use modeling to gain a deeper understanding of why the aspect ratio is important. In this way they provide valuable new insight into the factors controlling growth.

We thank the reviewer for the positive comments.

Besides the technological importance of this advance, the progress toward a fundamental understanding of mask-controlled growth is fascinating. Given the quality and broad interest of the work and the importance of the results, the paper seems highly appropriate for Nature Communications. There are some points in the results and discussion that could be clarified prior to publication, and these are addressed below.

We thank the reviewer for the positive and encouraging comments; the answers to all questions are given below.

Figure 2 shows "AFM images of the gallium droplets". Is the AFM performed in situ at the growth temperature? This should be stated explicitly. If the AFM is done at room temperature or even in air, it would still give useful insight, but in any case the reader should be informed how far these measurements directly reflect the growth conditions, since the wetting could change somewhat with temperature and atmosphere.

The AFM images are performed ex-situ at room temperature. We include this information in the revised manuscript (page 5, line 16).

The authors "claim here that it is rather the hole aspect ratio d/h that plays the universal role in producing the best nanowire seeds and hence achieving the high yield of nanowires." While the claim is adequately supported by the results, the wording seems a bit strong. The thickness ranges over only a factor of 2, while the aspect ratio of maximum yield varies by at least a factor of 1.5 (perhaps more, since we never see the curve for the 20nm film turn down). A more nuanced wording would seem appropriate, to reflect both the great advance in identifying the role of aspect ratio, and the remaining gaps in understanding and in achieving complete control.

We thank the referee for pointing this out. We have removed/modified the word 'universal' in the following way indicated in detail below:

→Abstract, line 4: We remove the word 'universal'

'Here, we propose a ~~universal~~ mechanism which gives a clear route for maximizing the nanowire yield in the self-catalyzed growth fashion.'

→Main text: page 3, line 8.

'We claim here that it is rather the hole aspect ratio d/h that plays a ~~universal~~ prominent role in producing the best nanowire seeds and hence achieving the high yield of nanowires.'

Does the theoretical modeling support the idea that hole aspect ratio is the key? Perhaps the takeaway message from the modeling could be made clearer for the non-expert reader. It seems that the modeling can be viewed as two quite separate pieces. The first involves the wetting of the Ga droplet. The key message here is that the hole should not have too wide/shallow an aspect ratio. If it does, the droplet will grow over the edge before it wets the entire bottom. This sets an upper limit on d/h , but it doesn't set any lower limit. The results shown using Surface Evolver do not actually illustrate this failure mode, but I hope I have understood correctly, because the general

principle seems simple and convincing. Since the wetting has no inherent length scale, it is truly the aspect ratio which matters here.

Theory indeed supports the experimental findings in the two ways: 1) showing that nucleation at the oxide-substrate line interface is preferred to any other location and 2) indicating that the droplet configuration depends on the hole aspect ratio. In turn, within the gallium droplet, this strongly influences diffusion/flux of the dissolved arsenic towards the oxide-substrate line interface.

We have modified the figures and the text to present more unambiguously the modeling results, in particular:

1. Clearer figures 2 and 4, highlighting the most important results of the modeling.
2. Clearer abstract.
3. Clearer concluding remarks describing the modeling results.

Then the puzzle becomes, why does a deep narrow hole not work? Here the modeling becomes more complex, and the inference more indirect. But the message seems to be that shadowing and As diffusion are causing failure for small d/h . The amount of shadowing is directly related to d/h , but diffusion is more complicated, so I'm not sure how directly this supports saying that d/h "plays the universal role". If the authors can offer any clarification or simple perspective, that would further improve the paper.

We agree with the referee that this point should have been elucidated in a clearer manner. We have now modified figure 4 so that the role of the droplet configuration in the arsenic gradient is clearer. For this, we add experimental data on the configuration of the droplet in the high and low yield situations. The link between these calculations and the high yield is now demonstrated more explicitly.

Directed flux toward the oxide-substrate interface line is consistent with the conditions leading to a high yield. We point out the reasons leading to a reduced yield for the smallest values of d/h , related to the sensitivity of the arsenic gradient to variations of the hole geometry and asymmetry.

*Changes, new text and figure from page 11:

In our machine, the molecular flux impinges on the surface at an angle of 45° . The flux into the gallium droplet is thus shadowed by the oxide mask at the interface with the substrate. The concentration of arsenic dissolved in the gallium droplet should thus be inhomogeneous, creating a gradient toward the substrate. The aspect ratio of the hole affects the shadowing and thus the direction of the gradient of dissolved arsenic in the droplet. Figure 4 (b) shows AFM measurements of gallium droplets filling 30 and 90 nm diameter holes in a 10 nm thick oxide mask, obtained after 10 min pre-deposition. These two configurations lead to the nanowire vertical yields of 65% and 3%, respectively (optimal yield is obtained for 45 nm holes, wetting configuration similar to 30 nm). Both droplets sit symmetrically in the nanoscale hole. However, for the small d/h ratio, the gallium droplet is pinned on top of the hole, exhibiting a contact angle close to 90° . For the high d/h ratio, the droplet is pinned at the bottom of the hole, wetting very little the sidewalls. We calculated the arsenic concentration profile for these configurations (the intermediate configurations are given in SI), by numerically solving Fick's equation, taking into account the sample rotation in MBE. Figure 4 and (c) depicts the steady state concentration profiles in droplets with the same d/h aspect ratio and configuration as measured by AFM. We present both the three-dimensional plot and a cut through the droplet center. Different droplet colors illustrate the gradient of the arsenic concentration and thus the direction of its diffusion, also indicated by arrows in the cross-sectional plots (right column of the figures). For low d/h ratios, the arsenic flux is directed towards the oxide-substrate interface line. This results in a strong shadowing effect in the holes with low d/h values. The lack of arsenic in these zones creates a permanent concentration gradient. By increasing the d/h ratio, the arsenic diffusion flux is turned outward from the substrate-oxide interface line. When the d/h ratios are too large, concentration gradient toward the oxide-substrate line

becomes negligible. Hence, the arsenic diffusion towards the substrate becomes homogeneous throughout the hole.

Let us now discuss the relevance of the effect using some numerical estimates. The diffusion coefficient of arsenic atoms in liquid gallium at the growth temperature was estimated at $2 \cdot 10^{-12} \text{ m}^2/\text{s}$. Therefore, the characteristic time for arsenic to diffuse through the droplet is about $5 \cdot 10^{-2} \text{ s}$. This is much smaller than the characteristic time of the single layer formation in the steady state nanowire growth ($\sim 1 \text{ s}$), while the nanowire nucleation delay is much longer and may take a few minutes. As expected, diffusion of arsenic cannot be the limiting factor for the nanowire growth or nucleation. However, it is expected that the diffusion profile influences the location of nanowire nucleation and in this way contributes to the vertical yield. For the symmetrical gradient toward the hole periphery, the liquid wets the hole's bottom edge and thus nucleation will occur there. Subsequent growth will cause this nucleus to cover the bottom edge, as discussed above. On the contrary, a reduced arsenic flux toward the oxide-substrate interface line or asymmetric diffusion profiles in the case of asymmetric hole filling lead to a higher degree of tilted growth. Finally, we address the decrease of the yield for the smallest d/h ratios. We believe that this decrease is mainly due to the difficulties in keeping symmetrical hole at small diameters. Given the strong gradient toward the oxide-substrate interface line at small d/h ratios, any inhomogeneity in the hole circular shape breaks the cylindrical symmetry of the flux. The decreased yield in this case is due to a higher arsenic gradient and its sensitivity to the hole asymmetry, which strongly increases for the smaller hole diameters (see SI 8). From this point of view, technical progress in nanofabrication should raise the yield for the smallest d/h values.

Figure 1. The diffusion model explaining the role of the hole aspect ratio d/h in improving the yield. a) top: schematics of the droplet configuration within the opening with respect to the impinging arsenic flux; bottom: schematics of the arsenic gradient created due to the sidewall shadowing. b) AFM measurements of the gallium droplets for 10 min pre-deposition time at an oxide thickness of 10 nm, showing the modification of the droplet shape from the full wetting, through partial wetting of the mask walls and ultimately non-wetting of the mask walls for largest d/h ratio; the top sketch showing the two wetting configurations in the cross-section c) three-dimensional representations and two-dimensional cuts of the calculated arsenic concentration profiles for the two wetting configurations corresponding to different d/h (top: $d/h=2$, bottom: $d/h=6$).

Some minor details:

I found Fig 2b a little confusing. I assume it is generated by Surface Evolver. Is the darker color a projection of the droplet volume, as one might intuitively expect by analogy with TEM? It looks more like a projection of the liquid-vapor interface; but if that is the case, it might be helpful to add a color for the liquid-oxide interface.

We have now slightly modified the figure and expanded the caption so that the content of figure 2 is clearer to the reader:

Figure 2. Filling of the nanoscale holes in SiO₂ mask on silicon. a) AFM images of the gallium droplets in a 45 nm diameter hole for three different pre-deposition times. Scale bar corresponds to 50 nm **b)** Illustrations of the evolution of the equilibrium droplet shape for increasing values of their volume. V_0 is the total volume of the hole. At small volumes, the droplet starts wetting asymmetrically from one side of the hole until the bottom interface is in full contact with the liquid phase. Further filling proceeds by increasing the wetting at the sidewalls. The morphologies were obtained using Surface Evolver for sessile droplets constrained in a cylindrical cavity with an aspect ratio of 4 (details in SI).

The paragraph describing the Surface Evolver calculations ends with a sentence "Complete filling of the holes leads to an improvement of the yield (with the highest values obtained within the optimal d/h from 4 to 6)." As the concluding sentence of a paragraph about modeling, it sounds as if this is a result of the modeling, but I don't think the authors meant to suggest this. A few extra words could clarify the distinction between model results and experimental results, along with the relation between the last sentence and the rest of the paragraph.

Indeed, it was not intention to sound as if this was the result of modeling. We have modified the text in the following way:

* Changes: page 6 last line:

'Intuitively, complete filling of the holes should lead to an improvement of the yield. One may think that this should naturally lead to layer-by-layer growth at the bottom of the hole.'

Reviewer #2:

This manuscript discusses self-catalyzed growth of GaAs nanowires on Si and in particular focuses on obtaining good nucleation and yield. There are interesting aspects and the work may be suitable for this journal. However I found it very difficult to read and am unsure what the authors' central claim is, even after reading it a few times. In order to be suitable for publication, this manuscript needs to be rewritten somewhat so that it is clear what the main claim or breakthrough is, and how this is supported by the experimental data. I suggest the authors focus on clarifying this in a revised manuscript.

We appreciate the positive comments by the reviewer. We have performed the following changes to render the main message of the manuscript clearer:

1. The abstract and conclusion has been partially rewritten, highlighting the role of the hole aspect ratio and symmetry for the achievement of high yield.
2. The parts on the surface evolver calculations and the diffusion of the dissolved arsenic are now more visual and specific.

We hope the reviewer will appreciate the new version.

To elaborate, the abstract gives the impression that positioning of nanowires on Si, or perhaps yield of positioned nanowires on Si, is the important breakthrough in the manuscript. However after reading the manuscript I have the impression that this in fact has been demonstrated before (in the section Optimal pattern design the authors indicate that similar results have been shown in Ref 7 for example). The abstract also seems to indicate that a new understanding of the nucleation is presented. But this seems to be a small part of the paper and appears to be rather speculative. The actual major result of the paper needs to be clarified here and throughout the paper.

We thank the referee for the interest in our work and results. The main finding is the fundamental understanding of the mechanisms leading to high yield. Only understanding can provide full reproducibility within and between groups and the basis for translation into other systems.

We are not aware of any other group that has discussed the yield as function of the aspect ratio and symmetry of the holes. In the cited publication, authors invoke an ideal oxide thickness. As we now know, thickness is not the only parameter and thus other groups had difficulties in reproducing these results. Here we demonstrate for the first time the key elements, which will allow all other groups to systematically obtain high yield nanowire arrays.

Some more specific points that should be addressed:

1. The abstract contains the sentence 'Contrary to common understanding, nucleation of nanowires occurs preferentially at the triple phase line between the liquid droplet...'. However it appears that preferential nucleation at the triple is exactly the current understanding in literature, so as written this sentence is factually incorrect. From the next sentence the authors appear to be distinguishing between the ring-like nucleation they discuss in the paper and more conventional nucleation followed by step flow, but this needs to be rephrased

We thank the reviewer for noticing this point. Indeed the sentence did not highlight correctly the findings that are against common understanding. We have now modified it.

New version of the abstract, in bold we highlight the relevant changes:

III-V semiconductor nanowires deterministically placed on top of silicon electronic platform would open many new avenues in the fields of silicon-based photonics, quantum technologies and energy harvesting. For this to become a reality, gold-free site-selected growth is necessary. Here, we propose a mechanism which gives a clear route for maximizing the nanowire yield in the self-catalyzed growth fashion. **It is widely accepted that growth of nanowires occurs on a layer-by-layer basis, starting at the triple-phase line. Contrary to common understanding, we find**

that vertical growth of nanowires starts at the oxide-substrate line interface, forming a ring-like structure several monolayers thick. This is granted by optimizing the diameter/height aspect ratio and cylindrical symmetry of holes, which impacts the diffusion flux of a group V element through the well-positioned group III droplet. This work provides new grounds for realistic integration of III-V semiconductor compounds on silicon and it opens new avenues for the organized growth of nanowires in a wide range of materials systems.

2. The introduction contains a paragraph discussing in-situ investigations of nucleation in nanowires. From this discussion I expected to see this topic appear again in the paper, but it doesn't appear to be related. It would help if this paragraph were given some context to how it relates to the present work, which doesn't seem to have anything to do with in-situ TEM.

We agree with the reviewer that we should explain why we mention the in situ techniques. In page 2 we explain that in situ studies of the initial stages of growth on a substrate are challenging and so far inexistent. In addition to this, we have added a sentence in page 8, as follows:

So far, in situ investigations of the initial stages of growth on a patterned substrate have not been achieved.

We also have added a very recent reference further demonstrating the layer-by-layer growth in nanowires at steady state: J.C. Harmand et al Phys. Rev. Lett. (2018), now ref 26.

3. As a general note, the authors refer to III-V nanowire growth in the title and abstract and claim that the results are general, but it appears that only GaAs has been studied. Is there any actual evidence that this work will apply to other III-V nanowires, which from my understanding are much more difficult to grow with this method? In the absence of direct evidence I think the claims of generality are exaggerated and should be reduced.

The results of this work can be used for the growth of nanowires using a low surface tension metal. Among the materials that have been obtained with these kinds of metals are: GaAs, GaSb, GaP, InAs, InP, Si, Ge, GeSn... some of these materials have never been attempted in an organized way. We are convinced that this work will inspire the groups working on all these families of materials. We now present these arguments in the abstract and conclusion.

*Abstract:

This work provides new grounds for realistic integration of III-V semiconductor compounds on silicon and opens new avenues for the organized growth of nanowires in a wide range of material systems.

*Conclusion:

They also provide novel insights on the formation of nanowires and can be translated to other material systems that can be grown with low surface energy catalysts such as GaSb, GaP, InAs, InP, GeSn, and Si.

Added references:

R.L. Woo et al Kinetic Control of Self-Catalyzed Indium Phosphide Nanowires, Nanocones, and Nanopillars, Nano Lett. 9, 2207 (2009)

L. Gao et al Self-Catalyzed Epitaxial Growth of Vertical Indium Phosphide Nanowires on Silicon Nano Lett. 9, 2223 (2009)

T. Xu et al, Type I band alignment in GaAs₈₁Sb₁₉/GaAs core-shell nanowires, Appl. Phys. Lett. 107, 112102 (2015)

Y. Xiang et al, Synthesis parameter space of bismuth catalyzed germanium nanowires, Appl. Phys. Lett. 94, 163101 (2009)

J. Tian et al Plasma-Assisted Growth of Silicon Nanowires by Sn Catalyst: Step-by-Step Observation, Nanoscale Res. Lett. 11, 455 (2016)

I. Zardo et al Gallium assisted plasma enhanced chemical vapor deposition of silicon nanowires. Nanotech. 20, 155602 (2009)

4. The observations of early stages of growth after removal of Ga droplets were very interesting and I would suggest the authors put more emphasis on this part of the paper. However I am not entirely convinced of the authors' claim that one sample is sufficient to see all stages of nucleation/early growth. The claim is that since nucleation occurs at different times for different holes, all stages can be seen simultaneously. However it could also be that the shapes evolve differently for different holes, and therefore I would argue that the authors should still have considered samples with several growth times, at least to prove their argument that their interpretation of the results is generally correct (and counter the impression that much of their model is just speculation).

We apologize if the manuscript gives the impression that only one sample was studied. In fact, many samples were measured at the different stages. We include this sample in the main manuscript as it shows features we repetitively see at different stages. This is due to the long and time-distributed incubation time. We clarify this now in the text.

*Modifications on page 7:

'These results are consistent among all samples investigated. Time-series have been conducted, and all the samples displaying growth times that are comparable to the incubation stage were showing identical behaviors.'

5. The last section, on diffusion in the droplet, appears to be purely a theoretical prediction. Can the authors comment on this, and how this could be tested experimentally (for example, would this model predict any trends that might be possible to observe?). Moreover, are there other possible explanations for the observed dependence on d/h of the holes, and how could these be tested against the proposed explanation?

These calculations were the only that could explain the results. We agree with the referee that this point should have been elucidated in a clearer manner. We emphasize this now in the manuscript. In particular, we have now modified figure 4 so that the role of the droplet configuration in the arsenic gradient is clearer. For this, we add experimental data on the configuration of the droplet in the high and low yield situations. The link between these calculations and the high yield is now more explicit.

Directed flux towards the oxide-substrate interface line is consistent with conditions leading to a high yield. We point out to the reasons leading to a reduction in the yield for the smallest values of d/h which is related to the sensitivity of the arsenic gradient from variations in the symmetry of the hole.

*Changes, new text and figure from page 11:

In our machine, the molecular flux impinges on the surface at an angle of 45° . The flux into the gallium droplet is thus shadowed by the oxide mask at the interface with the substrate. The concentration of arsenic dissolved in the gallium droplet should thus be inhomogeneous, creating a gradient toward the substrate. The aspect ratio of the hole affects the shadowing and thus the direction of the gradient of dissolved arsenic in the droplet. Figure 4 (b) shows AFM measurements of gallium droplets filling 30 and 90 nm diameter holes in a 10 nm thick oxide mask, obtained after 10 min pre-deposition. These two configurations lead to the nanowire vertical yields of 65% and 3%, respectively (optimal yield is obtained for 45 nm holes, wetting configuration similar to 30 nm). Both droplets sit symmetrically in the nanoscale hole. However, for the small d/h ratio, the gallium droplet is pinned on top of the hole, exhibiting a contact angle close to 90° . For the high d/h ratio, the droplet is pinned at the bottom of the hole, wetting very little the sidewalls. We calculated the arsenic concentration profile for these configurations (the intermediate configurations are given in SI), by numerically solving Fick's equation, taking into account the sample

rotation in MBE. Figure 4 and (c) depicts the steady state concentration profiles in droplets with the same d/h aspect ratio and configuration as measured by AFM. We present both the three-dimensional plot and a cut through the droplet center. Different droplet colors illustrate the gradient of the arsenic concentration and thus the direction of its diffusion, also indicated by arrows in the cross-sectional plots (right column of the figures). For low d/h ratios, the arsenic flux is directed towards the oxide-substrate interface line. This results in a strong shadowing effect in the holes with low d/h values. The lack of arsenic in these zones creates a permanent concentration gradient. By increasing the d/h ratio, the arsenic diffusion flux is turned outward from the substrate-oxide interface line. When the d/h ratios are too large, concentration gradient toward the oxide-substrate line becomes negligible. Hence, the arsenic diffusion towards the substrate becomes homogeneous throughout the hole.

Let us now discuss the relevance of the effect using some numerical estimates. The diffusion coefficient of arsenic atoms in liquid gallium at the growth temperature was estimated at $2 \cdot 10^{-12} \text{ m}^2/\text{s}$. Therefore, the characteristic time for arsenic to diffuse through the droplet is about $5 \cdot 10^{-2} \text{ s}$. This is much smaller than the characteristic time of the single layer formation in the steady state nanowire growth ($\sim 1 \text{ s}$), while the nanowire nucleation delay is much longer and may take a few minutes. As expected, diffusion of arsenic cannot be the limiting factor for the nanowire growth or nucleation. However, it is expected that the diffusion profile influences the location of nanowire nucleation and in this way contributes to the vertical yield. For the symmetrical gradient toward the hole periphery, the liquid wets the hole's bottom edge and thus nucleation will occur there. Subsequent growth will cause this nucleus to cover the bottom edge, as discussed above. On the contrary, a reduced arsenic flux toward the oxide-substrate interface line or asymmetric diffusion profiles in the case of asymmetric hole filling lead to a higher degree of tilted growth. Finally, we address the decrease of the yield for the smallest d/h ratios. We believe that this decrease is mainly due to the difficulties in keeping symmetrical hole at small diameters. Given the strong gradient toward the oxide-substrate interface line at small d/h ratios, any inhomogeneity in the hole circular shape breaks the cylindrical symmetry of the flux. The decreased yield in this case is due to a higher arsenic gradient and its sensitivity to the hole asymmetry, which strongly increases for the smaller hole diameters (see SI 8). From this point of view, technical progress in nanofabrication should raise the yield for the smallest d/h values.

Figure 3. The diffusion model explaining the role of the hole aspect ratio d/h in improving the yield. a) top: schematics of the droplet configuration within the opening with respect to the impinging arsenic flux; bottom: schematics of the arsenic gradient created due to the sidewall shadowing. b) AFM measurements of the gallium droplets for 10 min pre-deposition time at an oxide thickness of 10 nm, showing the modification of the droplet shape from the full wetting, through partial wetting of the mask walls and ultimately non-wetting of the mask walls for largest d/h ratio; the top sketch showing the two wetting configurations in the cross-section c) three-

dimensional representations and two-dimensional cuts of the calculated arsenic concentration profiles for the two wetting configurations corresponding to different d/h (*top: $d/h=2$, bottom: $d/h=6$*).

Reviewer #3:

This is a very interesting paper focusing on an important challenge of epitaxial deposition of a II-V semiconductor (GaAs) on a silicon substrate. To alleviate the problems of lattice strain and mismatch in interface polarity, GaAs nanowires are deposited onto the substrates in specific, targeted positions using patterned holes in a silicon oxide mask. The authors study how the size of the holes--specifically, the aspect ratio between the hole diameter and the height of the oxide--contributes to the orientation of the nanowire growth--whether the nanowires grow vertically, or at some other random angle. The experiments are very well-done and the data convincingly show that this hole aspect ratio, and the initial deposition of Ga into the holes, is what determines the orientation of the nanowires. The authors propose a model for this based on how the nanowires nucleate in the holes (i.e., at the edges of the holes if the Ga droplets are not completely filling the holes) and how As flux can also lead to diffusion gradients in the seeding droplet under certain conditions. I think that the experimental data are solid, and the proposed model seems convincing to me. I recommend publication of the paper with some additional minor comments:

We thank the reviewer for the positive and encouraging comments; we provide answers to the questions raised in the following.

1. In the abstract, defining that the aspect ratio is the hole diameter to the mask oxide thickness (line 19, pg 1). I had to read through the paper to figure out what aspect ratio was being referred to in the abstract, and since this is key to the paper, it would be useful to define it in the abstract.

We thank the referee for pointing this out. We have now included this definition in the abstract, in line 11. The abstract has also been rewritten in a clearer fashion:

III-V semiconductor nanowires deterministically placed on top of silicon electronic platform would open many new avenues in the fields of silicon-based photonics, quantum technologies and energy harvesting. For this to become a reality, gold-free site-selected growth is necessary. Here, we propose a mechanism, which gives a clear route for maximizing the nanowire yield in the self-catalyzed growth fashion. It is widely accepted that growth of nanowires occurs on a layer-by-layer basis, starting at the triple-phase line. **Contrary to common understanding, we find that vertical growth of nanowires starts at the mask-substrate line interface, forming a ring-like structure several monolayers thick. This is granted by optimizing the diameter/height aspect ratio and cylindrical symmetry of holes, which impacts the diffusion flux of a group V element through the well-positioned group III droplet.** This work provides new grounds for realistic integration of III-V semiconductor compounds on silicon and it opens new avenues for the organized growth of nanowires in a wide range of materials systems.

2. The authors refer to "layer-by-layer" growth and growth in a "multilayer" fashion often throughout the paper, yet never really define what layers they are referring to. I understand that researchers in the field will know, but the more general audience, even of materials researchers, probably will not know that they are referring to how the nanowires grow vertically from the substrate by the controlled addition of atomic layers... it would be helpful to have an explicit statement early in the paper about how the authors are defining layer-by-layer growth, etc.

We have now unified the terms and substituted multilayer and monolayer respectively by multiple and single layer. With this, we use only one term through the manuscript.

3. Pg 2, Line 43...the sentence about the absence of in situ TEM of nanowires growing from a substrate was confusing to me... at first, I thought that the paper was going to show in situ TEM data of the nanowire growth, but did not. I would revise this somehow to perhaps just say that nobody knows how to do such an in situ TEM

imaging, or something like that. At least clarifying that the paper is not going to show any in situ TEM data might be helpful to the reader.

We agree with the reviewer that we should explain why we mention the in situ techniques. We have added a sentence on page 8. We also include a recent reference on the latest results of in situ TEM work (nr 26?):

So far, in situ investigations at the initial stages of growth on a patterned substrate have not been achieved.

4. Related to the data in figure 1, I imagine that there is also a critical hole diameter that above which, nanowires do not grow epitaxially/vertically, regardless of the hole aspect ratio. Do the authors have some measure of this value, and can they state it for completeness? This is what the ref 3 is really about...the critical diameter needed to achieve a kind of defect-free growth. Also, is there a minimum diameter at which point the nanowires also do not grow? Or is the patterning dimension limited so this minimum hole size can't be reached? A comment about this lower limit would be useful as well.

Directed flux towards the oxide-substrate interface line is consistent with conditions leading to a high yield. This flux is suppressed for larger diameters, as shown by the calculations.

The reduction in the yield for the smallest values of d/h which is of a slightly different origin. In fact, it is related to the sensitivity of the arsenic gradient from variations in the symmetry of the hole at the smallest sizes (where geometry variations occur more frequently due to state-of-the-art technical limitations). This means that currently the minimum size is mostly due to technical reasons. We have added a comment on this point on page 13:

'Precise nanofabrication of more symmetrical holes in the oxide layer is expected to increase the yield for the smallest d/h values.'

5. Pg 5, line 91. Do the authors need to add a reference for the statement? "It has been shown that..." Or is this an introductory remark for the work that they are going to be talking about. It was confusing to me whether this was a sentence introducing the findings of this paper, or describing some previously published result.

We thank the referee for noticing this. Indeed, a link to the reference was missing here (nr 12). It is now included.

6. Figure 2. It would be helpful to have some labels placed on the images in (b). Are these snapshots at different times in the growth? Or with more Ga deposited? Also, the caption should state that those are computed shapes with Surface Evolver. This is stated in the text, but the caption offers little detail about what is shown in the image. Also, the scale of the scale bar in (a) is not provided. (Is it 45 nm?)

We have now added information on the surface evolver results on the figure and in the caption.

*Changes, page 7:

Figure 4. Filling of the nanoscale holes in SiO₂ mask on silicon. a) AFM images of the gallium droplets in a 45 nm diameter hole for three different pre-deposition times. Scale bar corresponds to 50 nm **b)** Illustrations of the evolution of the equilibrium droplet shape for increasing values of their volume. V_0 is the total volume of the hole. At small volumes, the droplet starts wetting asymmetrically from one side of the hole until the bottom interface is in full contact with the liquid phase. Further filling proceeds by increasing the wetting at the sidewalls. The morphologies were obtained using Surface Evolver for sessile droplets constrained in a cylindrical cavity with an aspect ratio of 4 (details in SI).

7. It would be helpful to add a sentence or two about the MBE deposition process in terms of how the flux is directional at 45 deg. Maybe provide a reference to a review paper or something like that? I'm more familiar with selective area CVD processes. In those cases, there is a higher diffusion flux at the boundaries of the hole than in the interior because of the hole geometry. Can something like this also happen in the MBE process or are the reactants somehow being accelerated to the substrate and impinging onto the substrate. A very brief explanation of this would be useful.

We thank the referee for this comment. Indeed one of the main characteristics of MBE is that the atomic or molecular precursors impinge to the substrate with a linear velocity from the source. Some adatoms may diffuse through the surface, but this is not the case of arsenic molecules. If they do not bond on the surface upon impingement, they subsequently desorb. We have now addressed this point and added the reference explaining the role of arsenic in MBE as well as a reference of a MBE book on page 11:

Figure 4 (a) illustrates the configuration of the gallium droplet in the symmetrically filled hole and the effect of the directionality of the As₄ flux in MBE.³² As adatom, arsenic does not diffuse on the surface. If it is not incorporated it is further desorbed.³³ In our machine, the molecular flux impinges on the surface at an angle of 45°. The arsenic flux into the gallium droplet is thus shadowed by the oxide mask at the interface with the substrate.

³² J.Y. Tsao 'Fundamentals of molecular beam epitaxy' Academic press (1993); ISBN: 9780127016252

³³ M.R. Ramdani, J.C. Harmand, F. Glas, G. Patriarche, L. Travers, 'Arsenic Pathways in Self-Catalyzed Growth of GaAs Nanowires' *Cryst. Growth Des.* 13, 91 (2013)

REVIEWERS' COMMENTS:

Reviewer #1 (Remarks to the Author):

I am satisfied with the author response to my previous comments and suggestions. In addition, the manuscript has been improved by changes made in response to the other reviewers. Therefore in my opinion the manuscript is now ready for publication in Nature Communications.

Reviewer #2 (Remarks to the Author):

The authors have substantially revised the manuscript according to all the comments. I find the revised manuscript much improved and suggest it is suitable for publication in this journal.

Reviewer #3 (Remarks to the Author):

The authors adequately addressed all of my concerns.